# Maternal Vaginal *Ureaplasma* spp. Colonization in Early Pregnancy Is Associated with Adverse Short- and Long-Term Outcome of Very Preterm Infants

**DOI:** 10.3390/children8040276

**Published:** 2021-04-03

**Authors:** Judith Rittenschober-Böhm, Tanja Habermüller, Thomas Waldhoer, Renate Fuiko, Stefan M. Schulz, Birgit Pimpel, Katharina Goeral, Armin Witt, Angelika Berger, Karin Pichler

**Affiliations:** 1Division of Neonatology, Intensive Care and Neuropediatrics, Department of Pediatrics and Adolescent Medicine, Comprehensive Center for Pediatrics, Medical University of Vienna, 1090 Vienna, Austria; t.habermueller@gmail.com (T.H.); renate.fuiko@muv.ac.at (R.F.); birgit.pimpel@muv.ac.at (B.P.); katharina.goeral@muv.ac.at (K.G.); angelika.berger@muv.ac.at (A.B.); Karin.a.pichler@muv.ac.at (K.P.); 2Department of Epidemiology, Center of Public Health, Medical University of Vienna, 1090 Vienna, Austria; thomas.waldhoer@muv.ac.at; 3Department of Pathology, Medical University of Vienna, 1090 Vienna, Austria; stefan.schulz@muv.ac.at; 4Division of Obstetrics and Feto-Maternal Medicine, Department of Obstetrics and Gynecology, Medical University of Vienna, 1090 Vienna, Austria; armin.witt@muv.ac.at

**Keywords:** intraventricular hemorrhage, neonatal short-term outcome, neurodevelopmental outcome, pregnancy, vaginal *Ureaplasma* spp. colonization

## Abstract

Vaginal colonization with *Ureaplasma* (*U.*) spp. has been shown to be associated with adverse pregnancy outcome; however, data on neonatal outcome are scarce. The aim of the study was to investigate whether maternal vaginal colonization with *U.* spp. in early pregnancy represents a risk factor for adverse short- or long-term outcome of preterm infants. Previously, 4330 pregnant women were enrolled in an observational multicenter study, analyzing the association between vaginal *U.* spp. colonization and spontaneous preterm birth. *U.* spp. colonization was diagnosed via PCR analysis from vaginal swabs. For this study, data on short-term outcome were collected from medical records and long-term outcome was examined via Bayley Scales of Infant Development at 24 months adjusted age. Two-hundred-and-thirty-eight children were born <33 weeks gestational age. After exclusion due to asphyxia, malformations, and lost-to-follow-up, data on short-term and long-term outcome were available from 222 and 92 infants, respectively. Results show a significant association between vaginal *U.* spp. colonization and severe intraventricular hemorrhage (10.4% vs. 2.6%, *p* = 0.03), retinopathy of prematurity (21.7% vs. 10.3%, *p* = 0.03), and adverse psychomotor outcome (24.3% vs. 1.8%, OR 13.154, 95%CI 1.6,110.2, *p* = 0.005). The data suggest an association between vaginal *U.* spp. colonization in early pregnancy and adverse short- and long-term outcome of very preterm infants.

## 1. Introduction

*Ureaplasma* (*U.*) spp. are the most common bacteria isolated from the amniotic cavity of women with preterm labor or preterm premature rupture of membranes (pPROM) [1,2,3]. The most frequent pathway of bacteria invading the intrauterine department is an ascending infection from the vagina [4]. A vaginal swab can be easily used as a screening method of *U.* spp. colonization during pregnancy. However, due to high vaginal *U.* spp. colonization rates of 40–80% in asymptomatic sexually active women, the clinical significance of vaginal *U.* spp. colonization in pregnancy remains uncertain [5,6,7]. In a previous multicenter study, we could demonstrate a significant association between vaginal *U.* spp. colonization in early pregnancy and an increased risk for spontaneous preterm birth. Furthermore we could show that the *Ureaplasma* biovar *U. parvum* rather than *U. urealyticum* was the more relevant biovar in this regard [8,9]. Similar results were found by other authors [6,10].

Less is known about the impact of vaginal *U.* spp. colonization on the outcome of neonates. Most studies investigating the association between *U.* spp. during pregnancy and morbidities of the preterm infant were based on the presence of *U.* spp. in the amniotic fluid or the neonatal respiratory tract [5,11,12,13,14]. Both, intrauterine and neonatal *U.* spp. colonization have been associated with several neonatal morbidities including bronchopulmonary dysplasia (BPD) and intraventricular hemorrhage (IVH) [5,7,11,12,15]. However, microbiological cultures from nose, throat, and tracheal aspirate performed in infants of vaginally colonized mothers immediately after birth have shown high transmission rates, ranging from 38% in term infants to 95% in very low birth weight (VLBW) infants [5,16]. This suggests that vaginal *U.* spp. colonization of the mother is likely followed by vertical transmission with subsequent colonization of the baby, particularly in preterm infants. Moreover, there is growing evidence that *U.* spp.-driven chorioamnionits can induce a fetal inflammatory response syndrome, which may contribute to neonatal injury [7,15].

Long-term outcome data after *U.* spp. colonization or infection in pregnancy are scarce and two existing studies revealed conflicting results concerning the impact of *U.* spp. on neurodevelopment [10,13].

In order to further evaluate the clinical significance of *U.* spp. colonization in early pregnancy, the aim of the present study was to investigate whether first-trimester vaginal colonization with *U.* spp. is associated with an adverse short- or long-term outcome of preterm infants born at <33 weeks of gestational age (wGA).

## 2. Materials and Methods

### 2.1. Study Design and Sample Collection

A total of 4330 pregnant women were enrolled in a prospective observational multicenter study during May 2008 to December 2012, as published before [8,9]. The aim of the study was to screen a large cohort of pregnant women for vaginal *U.* spp. colonization during routine nuchal translucency screening in order to investigate a potential association between vaginal *U.* spp. colonization in early pregnancy and an adverse pregnancy outcome. *U.* spp. colonization was diagnosed via DNA extraction from vaginal swabs and PCR analysis as described before [8,9]. Briefly, DNA was isolated from vaginal swabs with the QIAamp DNA Mini Kit (QIAGEN, Düsseldorf, Germany) and PCR reactions were run on a Mastercycler ep realplex4 S (Eppendorf, Hamburg, Germany) using primers and probes as previously described by Mallard et al. [17]. Results revealed a significant association between isolation of *U.* spp. and spontaneous preterm birth. In the current study, we focused on the association between vaginal *U.* spp. colonization and short- and long-term outcomes of very preterm infants. All infants delivered <33 wGA were included. Infants with asphyxia, fetal malformations, or chromosomal aberrations were excluded (Figure 1).

The study was approved by the ethics committees of the Medical University of Vienna (EK NR 655/2008) and the City of Vienna (EK 09-120-VK). All mothers gave written informed consent prior to participation.

### 2.2. Clinical Outcome

Data on vaginal colonization with *U.* spp. and pregnancy outcome were available from the database of the original study [9]. Since PCR analyses were performed for study purposes only, no macrolide eradication therapy was performed based on the results. Data on short- and long-term outcomes were obtained from patients’ electronic medical records. Early onset sepsis (EOS) was defined as a positive result from the blood culture taken immediately after birth plus clinical signs of infection. IVH was classified according to Papile et al. [18] and subgrouped as “severe IVH” if ≥ grade III was diagnosed. Periventricular leukomalacia (PVL) was defined as bilateral hyperechoic lesions observed in periventricular areas in both the coronal and parasagittal views on cranial ultrasound, subsequently evolving into at least one, and most frequently several, cystic lesions [19]. Necrotizing enterocolitis (NEC) was diagnosed if clinical and radiological signs consistent with ≥ grade II according to the staging system of Bell applied [20]. Screening for retinopathy of prematurity (ROP) was started at 5 weeks chronological age until vascularization was complete and staged according to the International Classification of Retinopathy of Prematurity [21]. Severe ROP was defined as ≥ grade III. BPD was defined as the need for any supplemental oxygen > 21% at 28 days of life (BPD 28 days) or 36 weeks of gestational age (BPD 36 weeks), respectively. Infants fulfilling the criteria of BDP 28 days without subsequent oxygen at 36 weeks of gestational age were not categorized as BDP 36 weeks.

### 2.3. Neurodevelopmental Follow-Up

The routine long-term follow-up program at our level III perinatal center includes a standardized test with the Bayley Scales of Infant Development at 24 months adjusted age for all preterm infants born <33 wGA, either using the Second or Third Edition (switch from Bayley II to Bayley III in August 2013) [22,23]. Mental Development Index (MDI) and Psychomotor Development Index (PDI) were obtained using Bayley II, Cognitive Composite Score (CCS), Language Composite Score (LCS), and Motor Composite Score (MCS) using Bayley III. For comparison, the following conversions were used: “MDI = (CCS+LCS)/2” and “PDI=MCS”. MDI and PDI were used as outcome parameters. Scores of 100 ± 15 represent the mean ± standard deviation (SD). Patients were classified according to their scores as normal (≥85), mildly impaired (70–84, <1SD), or severely impaired (<70, <2SD). German norms of the Bayley III were used [24]. Bayley testing was not performed in infants transferred from our center to peripheral hospitals before discharge or infants born in one of the two other hospitals participating in the multicenter study.

### 2.4. Statistical Analysis

Mean, SD, and percentage were used to describe the sample. GA and birth weight between *U.* spp.-positive and -negative patients were compared by means of the unpaired t-test. The effect of vaginal colonization with *U.* spp. was estimated using odds ratios (OR) and corresponding 95% confidence intervals (CI). Significance level for statistical tests was set to 5%. *p*-values are not adjusted for multiple testing and are to be interpreted exploratorily only. Statistical analysis was done using SAS version 9.4 and MS Office Excel.

## 3. Results

### 3.1. Study Population

A total of 159 women delivered at <33 wGA (89 singletons, 62 twins, 7 triplets, 1 quadruplet), resulting in a total of 238 preterm infants eligible for the study. Eighty women were negative and 79 were positive for vaginal *U.* spp. colonization. The high rate of multiple births in our study group is typical for our perinatal center and explained by the fact that it is the only level III perinatal center caring for high-risk pregnancies in eastern Austria and therefore high-risk multiple pregnancies are preferentially delivered here.

Of the 238 study patients, 8 had to be excluded because of severe asphyxia, fetal malformation, or chromosomal aberration, and 8 babies born in peripheral hospitals were lost for follow up, leaving 222 infants (149 pregnancies) for analysis of short-term outcomes (Figure 1). Of those, 116 babies (52.3%) were born to mothers negative for vaginal *U.* spp. colonization and 106 (47.7%) to mothers with vaginal *U.* spp. colonization (93 positive for *U. parvum*, 9 for *U. urealyticum*, 4 for both). Sixteen patients died during the course of the study and 114 patients had no routine Bayley assessment at 24 months adjusted age (80 due to being transferred to or born in peripheral hospitals without Bayley assessment, 34 lost for follow up), leaving 92 patients for analysis of long-term outcomes (Figure 1). Clinical characteristics of study patients according to vaginal *U.* spp. colonization are given in Table 1. There were no significant differences regarding birthweight, wGA, sex, antenatal steroids, pPROM, or percentage of singleton pregnancies between groups.

### 3.2. Short-Term Outcome

The short-term outcome of study patients in relation to vaginal *U.* spp. colonization status of the mother is summarized in Table 2. Significantly more children were diagnosed with severe IVH (10.4% vs. 2.6%, *p* = 0.03) or any ROP (21.7% vs. 10.3% *p* = 0.03) in the *U.* spp.-positive group compared to the *U.* spp.-negative group. More infants in the *U.* spp.-positive group suffered from severe ROP; however, this difference did not reach statistical difference. This might be attributed to the overall small number of patients with severe ROP. No significant difference was found between groups in the rates of EOS, PVL, NEC, BPD 28 days, BPD 36 weeks, and death.

A subgroup analysis of *U.* spp. biovars (*U. parvum, U. urealyticum*) did not show any further associations, noting that only nine preterm infants were born to mothers positive for *U. urealyticum*.

Short-term outcome data separated for children with long-term outcome examination at 24 months adjusted age are given in Appendix A. Children seen in our follow-up clinic had, in general, higher rates of adverse short-term outcome compared to those lost-to-follow-up (who were more likely to be transferred to peripheral hospitals before discharge). In this subgroup of preterm infants with long-term outcome data, a statistically significant difference between the *U.* spp.-positive group and the *U.* spp.-negative group was found for severe and any ROP as well as for any IVH and BPD at 28 days.

### 3.3. Long-Term Outcome

Standardized Bayley test results at 24 month adjusted age were available from 92 infants, of whom 55 (59.8%) were born to mothers negative for vaginal *U.* spp. colonization and 37 (40.2%) to mothers with vaginal *U.* spp. colonization.

Psychomotor and Mental Development Indices are given in Table 3. Overall, psychomotor examination showed normal scores in 64 infants (69.6%); 18 children (19.6%) presented with mild and 10 (10.9%) with severe abnormalities. Regarding the Mental Developmental Index, 45 infants (48.9%) were classified as normal, 23 (25.0%) had mild, and 24 (26.1%) severe abnormalities. Of 10 and 24 infants with severe PDI and MDI scores, 3 and 4, respectively, were suffering from severe IVH.

Infants in the *U.* spp.-positive group had a significantly higher risk for a severely abnormal psychomotor outcome (24.3% vs. 1.8%, OR 13.15 (95% CI 1.6,110.2), *p* = 0.005) compared to the *U.* spp.-negative group. While the rate of children with normal PDI test results was similar in both groups, babies born to mothers with vaginal *U.* spp. colonization presented more often with severe abnormalities and less often with mild abnormalities compared to the *U.* spp.-negative group. There was no significant difference between groups in mental outcome, although the rate of children presenting with severe abnormalities in the mental score was slightly higher in the *U.* spp.-positive group (OR 1.81 (95% CI 0.7,5.0)).

In order to adjust the psychomotor outcome by severe IVH, a known risk factor for adverse neurodevelopmental outcome, given the small sample size, outcome data were analyzed in subgroups (severe IVH yes or no). While the association of having a higher risk for a severely abnormal psychomotor outcome with maternal vaginal *U.* spp colonization was significant in the group of children without severe IVH (*n* = 86; *p* = 0.001), there was no significant difference in the group of children suffering from severe IVH (*n* = 6; *p* = 0.99). There was no significant difference in both subgroups when adjusting the mental outcome by severe IVH.

## 4. Discussion

While the clinical significance of vaginal *U.* spp. colonization was controversially discussed for a long time, there is growing evidence demonstrating a potential adverse impact of these commensals of the vaginal microbiome on pregnancy and neonatal outcome [5,7,8,9]. In a recent review, Silwedel et al. concluded that “the profound relevance of *Ureaplasma* colonization on preterm infants may still be underestimated” [7]. The present study reports data on short- and long-term outcome of preterm infants after maternal vaginal *U.* spp. colonization in early pregnancy, suggesting that, not only intrauterine *U.* spp. infection, but also vaginal colonization with *U.* spp. might be associated with an adverse neurodevelopmental outcome.

These data add to earlier reports on the short-term impact of maternal vaginal *U.* spp. colonization on preterm infants. Abele-Horn et al. as well as Kafetzis et al. cultured nose, throat, and tracheal aspirate from infants of vaginally colonized mothers after birth and found high transmission rates, ranging from 38% in term infants to 95% in VLBW infants [5,16]. Abele-Horn et al. reported that VLBW infants with *U.* spp. colonization of the respiratory tract were at higher risk for respiratory distress syndrome, IVH, and BPD [5]. Similarly, Kafetzis et al. found the highest rate of transmission and respiratory tract colonization with *U.* spp. in infants < 1500 g and these children had a significantly increased risk for the development of BPD and mortality compared to non-colonized infants [16]. On the contrary, a small study by Suzuki et al. did not find any association between vaginal *U.* spp. colonization after pPROM and neonatal outcome [10].

In contrast to the earlier mentioned studies, we analyzed the impact of first-trimester vaginal colonization of mothers on neonatal short- and long-term outcome and found a significant association between *U.* spp. colonization and severe IVH as well as any ROP. Moreover, we report long-term neurodevelopmental outcomes of preterm infants after vaginal *U.* spp. colonization of the mother in early pregnancy, indicating for the first time a possible association between vaginal *U.* spp. colonization and an adverse psychomotor outcome of the infant at 24 months corrected age.

As stated before, the most common pathway of bacteria invading the intra-amniotic compartment is an ascending infection from the vagina [4]. For a long time, amniotic fluid and amniotic membranes were regarded as sterile compartment with cervix, cervical mucus and amniotic membranes acting as barriers for ascending bacteria [25]. However, more recent data suggest that bacterial invasion of the amniotic cavity can occur without pPROM. According to a model proposed by Kim et al., a restricted cervical region of the chorioamnion allows organisms to invade the uterine cavity, where they proliferate in the amniotic fluid and subsequently re-invade the chorioamniotic membranes [25]. This has also been shown for Ureaplasmas: By analyzing amniotic fluid from mid-trimester amniocenteses, Cassell et al. [26] as well as Gray et al. [27] demonstrated that *U.* spp can invade the amniotic fluid despite intact membranes and tend to stay clinically silent for several weeks; however, it can lead to chronic infection and chorioamnionitis. Once Ureaplasmas have entered the amniotic fluid and choriodecidual tissue, they can activate the decidua and fetal membranes to produce a number of different cytokines and subsequently modulate the release of pro-inflammatory cytokines and inflammatory mediators [28,29,30]. This *U.* spp driven host immune response might affect the fetus and contribute to the pathogenesis of neonatal injury [7,30].

Usually, U. spp infections are limited to mucosal surfaces [31]. However, in the case of limited host defense, as in the developing fetus, they can also disseminate into organs [30,31]. An intrauterine immune response may therefore also be induced by the fetus, called fetal inflammatory response syndrome (FIRS), which is a risk factor for severe neonatal morbidities [32].

The association between *U.* spp. and IVH has been reported before. Viscardi et al. obtained cord blood or venous blood from VLBW infants within 12h after delivery and found that children with serum PCR positive for *U.* spp. had a 2.3-fold higher risk for severe IVH compared to patients with negative PCR results [15]. Similar data were presented by Resch et al. who found that preterm infants with tracheal *U.* spp. colonization had significantly increased rates of IVH and seizures [12]. A previous study of our own group on the neonatal outcome after isolation of *U.* spp. from the amniotic cavity at birth also showed a significantly increased risk for IVH and BPD [11]. Data of the current study also suggest that maternal vaginal *U.* spp. colonization could be associated with an increased risk for severe IVH of very preterm infants.

Moreover, the study by Viscardi et al. showed that *U.* spp. not only colonize the respiratory tract but can also invade the bloodstream, cross the immature blood-brain barrier, and are associated with systemic and central nervous system inflammation. In an earlier study, Viscardi et al. demonstrated that elevated inflammatory cytokines in serum or cerebrospinal fluid were associated with higher rates of IVH or white matter lesions, designating it the “cytokine hypothesis of brain injury” [33]. These data are supported by a meta-analysis by Huang et al., reporting that antenatal infection increased the incidence of IVH in preterm infants [34].

Additionally, Ureaplasmas have been shown to effect cytokine levels and tend to shift the balance towards pro-inflammation [35,36]. Neonatal organ injury may therefore be based on a robust intrauterine immune response and result from cytokine associated tissue damage as well as higher host vulnerability to secondary injuries such as postnatal infections [7,15,33].

Similarly, the *Ureaplasma*-triggered proinflammatory response might be involved in the pathogenesis of ROP. The association between *U.* spp. colonization of preterm infants and ROP was earlier described by Ozdemir et al. [14] and now confirmed by data of our study. An increased vascular endothelial growth factor (VEGF) release by macrophages exposed to *U.* spp. and subsequent modulation of angiogenesis might play a critical role [14,37].

Long-term outcome after *U.* spp. colonization has not been adequately studied so far. To our best knowledge, only two studies investigated long-term outcome after intrauterine or vaginal *U.* spp colonization of the mothers, reporting contradictory findings [10,13]. Results of this study confirm published data of our research group showing an association between intrauterine *U.* spp. at birth and adverse psychomotor outcome at 24 months adjusted age [13]. On the contrary, a small Japanese study did not find any association between vaginal *U.* spp. colonization and long-term outcome [10]. However, it seems plausible that *U.* spp.-associated chronic inflammation and subsequent immune response can contribute to multifactorial brain injury, resulting in white matter disease, which is the most important risk factor for an adverse neurological outcome [13,38]. In a mouse model of *U.* spp. induced perinatal infection, Normann et al. demonstrated that antenatal exposure to *U.* spp. can provoke central microgliosis, delayed myelination, and disturbed brain development by decreasing the number auf neurons in the neocortex. Similarly, Kelleher et al. examined preterm rhesus macaques after intrauterine *U. parvum* infection via postnatal brain magnetic resonance imaging, suggesting potential perturbation of brain growth and white matter maturation [39]. These disturbances of brain development may participate in the process leading to potential adverse long-term neurodevelopmental outcomes of *U.* spp. exposed infants [40]. In our cohort, the rate of infants with normal PDI tests was similar in both groups, while babies of *U.* spp.-positive mothers significantly more often presented with severe neuromotor abnormalities. It can only be speculated that this could be attributed to the fact that Ureaplasmas stay clinically silent in many cases while causing chronic and prolonged inflammation and a robust host immune response with subsequent multifactorial brain injury in others [7].

We would have also expected a potential association between *U.* spp. colonization and BPD of preterm infants. Animal models as well as human studies have shown that the *Ureaplasma*-triggered proinflammatory response in the lungs might affect alveolar development and promote abnormal septation and interstitial fibrosis, both characteristics of BPD [41]. We speculate that the lack of association has to be attributed to the low number of study patients, which is a clear limitation of our study. Preterm infants included in this study derived from more than 4000 pregnancies included in the original multicenter study, which demonstrates the difficulty in obtaining a study group large enough to satisfactorily answer these questions with enough statistical power. As a consequence, results on short- and long-term outcome were not adjusted for GA or birth weight. Estimating interaction terms in statistical models or analogously sub-grouping needs a larger sample size. However, as shown in Table 1, groups were comparable in terms of GA and other clinical characteristics.

Moreover, it has to be acknowledged that data on long-term outcomes might be limited by an ascertainment bias, as the group of infants transferred to peripheral hospitals with no Bayley testing generally had a less complicated postnatal course compared to those staying at the level III hospital, implicating that results on long-term outcome refer to a group of infants with more severe disease.

Another limitation of the study is that no data were available on maternal, fetal, or neonatal inflammatory parameters, nor any data on amniotic or neonatal *U.* spp. colonization around birth. However, the high risk of vertical transmission with subsequent colonization of the baby after *U.* spp. colonization of the mother is well documented in two other studies [5,16].

## 5. Conclusions

In conclusion, results of our study suggest that vaginal *U.* spp. colonization in early pregnancy might increase the risk for adverse short- and long-term outcomes of preterm infants including an adverse psychomotor outcome at 24 months adjusted age. Whether early screening and treatment of colonized mothers can prevent adverse outcomes in preterm infants needs to be tested in future studies.

## Figures and Tables

**Figure 1 children-08-00276-f001:**
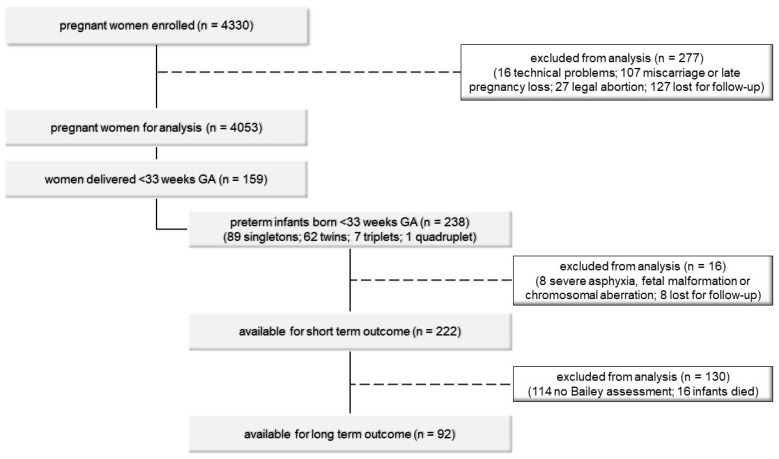
Patient flow chart.

**Table 1 children-08-00276-t001:** Clinical characteristics of study patients according to vaginal Ureaplasma spp. colonization.

	Whole Cohort *n* = 222	*Ureaplasma* spp. Negative *n* = 116	*Ureaplasma* spp. Positive *n* = 106	*p*-Value
Birthweight, g mean (SD)	1264 (432)	1302 (400)	1222 (461)	0.17
Gestational age, weeks mean (SD)	29.4 (2.6)	29.6 (2.4)	29.1 (2.8)	0.23
Male sex, %	55.4	59.5	50.9	0.23
Antenatal steroids any, %	93.2	91.2	95.2	0.29
pPROM, %	40.1	38.8	41.5	0.68
Singleton pregnancy, %	36.9	31.0	43.4	0.06

**Table 2 children-08-00276-t002:** Short-term clinical outcome of study patients according to vaginal Ureaplasma spp. colonization.

	Whole Cohort *n* = 222	*Ureaplasma* spp. Negative *n* = 116	*Ureaplasma* spp. Positive *n* = 106	*p*-Value
EOS, *n* (%)	4 (1.8)	2 (1.7)	2 (1.8)	0.99
Any IVH, *n* (%)	38 (17.1)	18 (15.5)	20 (18.9)	0.59
Severe IVH, n (%)	14 (6.3)	3 (2.6)	11 (10.4)	0.03
PVL, *n* (%)	6 (2.7)	2 (1.7)	4 (3.8)	0.43
NEC, *n* (%)	13 (5.9)	4 (3.4)	9 (8.5)	0.15
Any ROP, *n* (%)	35 (15.8)	12 (10.3)	23 (21.7)	0.03
Severe ROP, *n* (%)	9 (4.1)	3 (2.6)	6 (5.7)	0.32
BPD 28 days #, *n* (%)	59 (26.6)	27 (23.3)	32 (30.2)	0.29
BPD 36 weeks #, *n* (%)	17 (7.7)	10 (8.6)	7 (6.6)	0.62
Death, *n* (%)	16 (7.2)	5 (4.3)	11 (10.4)	0.12

# only patients surviving 28 days of life and 36 weeks of adjusted age, respectively, are included.

**Table 3 children-08-00276-t003:** Neurodevelopmental outcome at 24 months adjusted age in relation to vaginal *Ureaplasma* spp. colonization status of the mother. Odds ratios apply to comparison with infants with normal scores as reference group. Motor outcome represented as Bayley Mental Development Index Scores, Mental outcome represented as Bayley Psychomotor Development Index scores.

	*Ureaplasma* spp. Negative *n* = 55	*Ureaplasma* spp. Positive *n* = 37	OR (CI), *p*-Value
Motor outcome			
Normal (≥85), *n* (%)	38 (69.1)	26 (70.3)	
Mildly abnormal (70–84), *n* (%)	16 (29.1)	2 (5.4)	0.18 (0.04,0.86) *p* = 0.02
Severely abnormal (<70), *n* (%)	1 (1.8)	9 (24.3)	13.15 (1.57,110.18) *p* = 0.005
Mental outcome			
Normal (≥85), *n* (%)	29 (52.7)	16 (43.2)	
Mildly abnormal (70–84), *n* (%)	14 (25.5)	9 (24.3)	1.17 (0.41,3.28) *p* = 0.8
Severely abnormal (<70), *n* (%)	12 (21.8)	12 (32.4)	1.81 (0.66,4.96) *p* = 0.31

## Data Availability

All data requests should be submitted to the corresponding author for consideration. Access to anonymized data may be granted following review.

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
