# Peer review of "Maternal Vaginal Ureaplasma spp. Colonization in Early Pregnancy Is Associated with Adverse Short- and Long-Term Outcome of Very Preterm Infants"

_children, 2021, doi:10.3390/children8040276_

Round 1

Reviewer 1 Report

Major comments:

When I compared the frequencies of infantile diseases between Table 2 and Supplementary Figure S1, the frequencies of all diseases (and subcategories) were higher in Figure S1 than in Table 2, suggesting that the cohort of long-term observation would have had bias toward to more severe morbidity. If infants with less severe diseases were lost of follow-up, the results for long-term outcomes may reflect the infants with severe diseases. However, this study, for the first time, showed the possible association between vaginal Ureaplasma colonization in the first trimester and the increase of frequency of severe IVH, any ROP, and severely abnormal motor outcome. The results will attract the Obstetrician the measurement of presence/absence of Ureaplasma spp.

Minor comments:

  1. Abstract: Line 28-30: As for adverse psychomotor outcome, both OR (95%CI) and frequency (%) should be shown.
  2. Methods: Line 75-76: To guarantee a reproducibility of study, would you add the detail of PCR method (primer sets for Ureaplasma spp.).
  3. Methods: The numbers of BPD36 were lower than the numbers of BPD28, suggesting that the BPD36 in the current study included only moderate and severe BPD36 alone. Therefore, you should rewrite the definition of BPD28 and BPD 36 more detail in the current manuscript.
  4. Results: Although you showed the association between Ureaplasma spp. Colonization in the first trimester and the increase of the frequency of severe IVH, any ROP, and severely abnormal motor outcome, you did not show the intermediate variables between them. In the current knowledge, ureaplasma spp. is associated with intrauterine infection, histological chorioamnionitis (CAM), and/or intra-respiratory tract colonization. However, you did not show such intermediate variables in this study. If possible, I demand you to add at least presence/absence of CAM.
  5. Discussion: You should discuss the reasons why vaginal Ureaplasma spp. colonization could be resulted in the occurrence of severe IVH, any ROP, and severely abnormal motor outcome. You should discuss for the intermediate variables, such as intrauterine infection, fetal infection, CAM, and/or infantile Ureaplasma spp. colonization, quoting appropriate references.

Reviewer 2 Report

Intrauterine Ureaplasma infection is associated with preterm birth; however, the clinical importance of vaginal colonization of Ureaplasma is still unclear. Rittenschober-Bohm showed that vaginal Ureaplasma colonization was associated with severe IVH, retinopathy, and severe psychomotor complications. 1. There are more multiple births than single births among 159 women delivered at <=32 wGA. Please describe the reason in the result section. And if you consider among only single deliveries, will the risk of BPD increase? Please discuss. 2. There is a mixed description of < 33 weeks and <= 32 weeks; please unify. 3. Please describe whether there is a difference in the cause of IVH due to vaginal delivery or cesarean section. And if the IVH is related to either vaginal delivery or cesarean section, discuss which is considered to be more influential, Ureaplasma, or differences in the way of delivery. 4. Table 3 shows that the motor outcome of severe disease is increased by Ureaplasma colonization, while that of mild patients is significantly higher in the absence of Ureaplasma. Explain this discrepancy. Minor point 5. Line 144, delete “our.” It is duplicated.

Reviewer 3 Report

  1. Was the placental pathology unavailable? The information on placental pathology would give more discussion point.
  2. The authors did not multivariate analysis. Even though, there are no significant differences in baseline demographics (gestational age, birth weight, gender), stratification (subgroup analysis) by gestational age and birth weight might give more insight into the association of maternal Ureaplama spp. colonization and neonatal outcomes.
  3. The authors did not adjust long-term neurodevelopmental outcomes by short-term outcomes. Alt least, long-term neurodevelopmental outcomes should be adjusted by severe IVH.
  4. Sepsis and periventricular leukomalacia are is one of the risk factors for adverse neurological outcome. Can you further include these two variables in analysis?   

Round 2

Reviewer 1 Report

Although the fact that the authors did not collect the information of histological CAM, the current study showed the important correlation of first trimester Ureaplasma and the infantile long-term prognosis, for the first time. In the future study, the relationship among first trimester Ureaplasma, intermediate variables including CAM, and the neurodevelopmental outcomes. The revision was appropriately performed.

Reviewer 3 Report

The authors responded well to my concerns and advice. The manuscript was revised appropriately. I have no further comment for this manuscript.